# Learning Graph Search Heuristics

**Michal Pándy**[*]
University of Cambridge

**Weikang Qiu**
Yale University

**Gabriele Corso**
MIT

**Petar Veličković**
DeepMind

**Rex Ying**
Yale University

**Jure Leskovec**
Stanford University

**Pietro Liò**
University of Cambridge

## Abstract

Searching for a path between two nodes in a graph is one of the most well-studied and fundamental problems in computer science. In numerous domains such as robotics, AI, or biology, practitioners develop search heuristics to accelerate their pathfinding algorithms. However, it is a laborious and complex process to hand-design heuristics based on the problem and the structure of a given use case. Here we present PHIL (Path Heuristic with Imitation Learning), a novel neural architecture and a training algorithm for discovering graph search and navigation heuristics from data by leveraging recent advances in imitation learning and graph representation learning. At training time, we aggregate datasets of search trajectories and ground-truth shortest path distances, which we use to train a specialized graph neural network-based heuristic function using backpropagation through steps of the pathfinding process. Our heuristic function learns graph embeddings useful for inferring node distances, runs in constant time independent of graph sizes, and can be easily incorporated in an algorithm such as A* at test time. Experiments show that PHIL reduces the number of explored nodes compared to state-of-the-art methods on benchmark datasets by $58.5\%$ on average, can be directly applied in diverse graphs ranging from biological networks to road networks, and allows for fast planning in time-critical robotics domains.

## 1 Introduction

Search heuristics are essential in several domains, including robotics, AI, biology, and chemistry [1–6]. For example, in robotics, complex robot geometries often yield slow collision checks, and search algorithms are constrained by the robot's onboard computation resources, requiring well-performing search heuristics that visit as few nodes as possible [1, 4]. In AI, domain-specific search heuristics are useful for improving the performance of inference engines operating on knowledge bases [3, 5]. Search heuristics have been previously also developed to reduce search efforts in protein-protein interaction networks [6] and in planning chemical reactions that can synthesize target chemical products [2]. This broad set of applications underlines the importance of good search heuristics that are applicable to a wide range of problems.

The search task can be formulated as a pathfinding problem on a graph, where given a graph, the task is to navigate and find a short feasible path from a start node to a goal node, while in the process visiting as few nodes as possible (Figure 1). The most straightforward approach would be to launch a search algorithm such as breadth-first search (BFS) and iteratively expand the graph from the start node until it reaches the goal node. Since BFS does not harness any prior knowledge about the graph, it usually visits many nodes before reaching the goal, which is expensive in cases such as robotics where visiting nodes is costly. To visit fewer nodes during the search, one may use domain-specific information about the graph via a *heuristic* function [7], which allows one to define a distance metric on graph nodes to prune directions that seem less promising to explore. Unfortunately, coming up with good search heuristics requires significant domain expertise and manual effort.

---

[*]Correspondence to michalpandy@google.com.

M. Pándy et al., Learning Graph Search Heuristics. *Proceedings of the First Learning on Graphs Conference (LoG 2022)*, PMLR 198, Virtual Event, December 9–12, 2022.

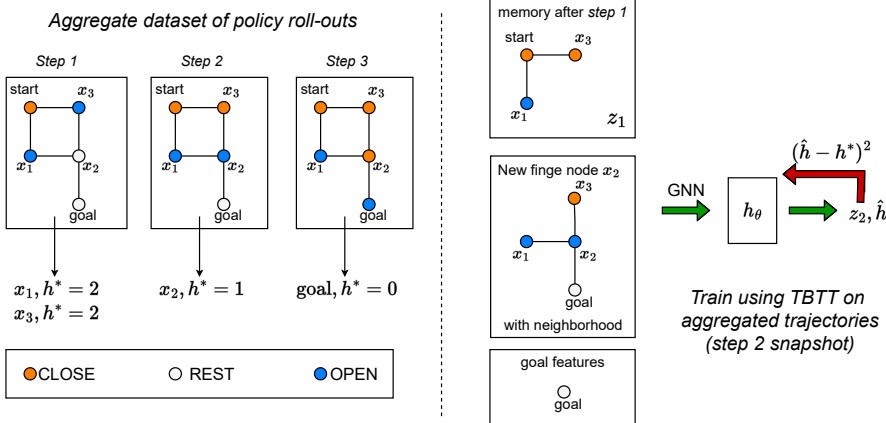

**Figure 1:** The goal is to navigate (find a path) from the start to the goal node. While BFS visits many nodes to find a start-to-goal path (left), one can use a heuristic based on the features of the nodes (e.g., Euclidean distance) on the graph to reduce the search effort (middle). We propose PHIL to learn a tailored search heuristic for a given graph, capable of reducing the number of visited nodes even further by exploiting the inductive biases of the graph (right).

While there has been significant progress in designing search heuristics, it remains a challenging problem. Classical approaches [8, 9] tend to hand-design search heuristics, which requires domain knowledge and a lot of trial and error. To alleviate this problem, there has been significant development in general-purpose search heuristics based on trading-off greedy expansions and novelty-based exploration [10–13] or search problem simplifications [14–16]. These approaches alleviate some of the common pitfalls of goal-directed heuristics, but we demonstrate that if possible, it is useful to learn domain-specific heuristics that can better exploit problem structure.

On the other hand, learning-based methods face a set of different challenges. Firstly, the data distribution is not i.i.d., as newly encountered graph nodes depend on past heuristic values, which means that supervised learning-based methods are not directly applicable. Secondly, heuristics should run fast, with ideally constant time complexity. Otherwise, the overall asymptotic time complexity of the search procedure could be increased. Finally, as the environment (search graph) sizes increase, reinforcement learning-based heuristic learning approaches tend to perform poorly [1]. State-of-the-art imitation learning-based methods can learn useful search heuristics [1]; however, these methods still rely on feature-engineering for a specific domain and do not generally guarantee a constant time complexity with respect to graph sizes.

**Figure 2:** Main components of PHIL: On the left, using a greedy mixture policy induced by the current version of our parameterized heuristic $h_\theta$ and an oracle heuristic $h^*$ (i.e., a heuristic that correctly determines distances between nodes), we roll-out a search trajectory from the start node to the goal node. Each trajectory step contains a set of newly added fringe nodes with bounded random subsets of their 1-hop neighborhoods and their oracle ($h^*$) distances to the goal node. Trajectories are aggregated throughout the training procedure. On the right, we use truncated backpropagation through time on each collected trajectory to train $h_\theta$, where $\hat{h}$ is the predicted distance between $x_2$ and $x_g$, and $z_2$ is the updated state of the memory. Here, the memory captures the embedding of the graph visited so far.

In this paper, we propose *Path Heuristic with Imitation Learning* (PHIL), a framework that extends the recent imitation learning-based heuristic search paradigm with a learnable *explored graph memory*. This means that PHIL learns a representation that allows it to capture the structure of the so far explored graph, so that it can then better select what node to explore next (Figure 2). We train our approach to predict the node-to-goal distances ($h^*$ in Figure 2) of graph nodes during search. To train our memory module, which captures the explored graph, we use truncated backpropagation through time (TBTT) [17], where we utilize ground-truth node-to-goal distances as a supervision signal at each search step. Our TBTT procedure is embedded within an adaptation of the AggreVaTe imitation learning algorithm [18]. PHIL also includes a *specialized graph neural network architecture*, which allows us to apply PHIL to diverse graphs from different domains.

We evaluate PHIL on standard benchmark heuristic learning datasets (Section 5.1), diverse graph-based datasets from different domains (Section 5.2), and practical UAV flight use cases (Section 5.3). Experiments demonstrate that PHIL outperforms state-of-the-art heuristic learning methods up to $4\times$. Further, PHIL performs within $4.9\%$ of an oracle in indoor drone planning scenarios, which is up to a $21.5\%$ reduction compared with commonly used approaches. In practice, our contributions enable practitioners to quickly extract useful search heuristics from their graph datasets without any hand-engineering.

## 2 Preliminaries

**Graph search**. Suppose that we are given an unweighted connected graph $\mathcal{G} = (\mathcal{V}, \mathcal{E})$, where $\mathcal{V}$ is a set of nodes, and $\mathcal{E}$ a corresponding set of edges. Further suppose that each node $i \in \mathcal{V}$ has corresponding features $x_i \in \mathbb{R}^{D_v}$, and each edge $(i, j) \in \mathcal{E}$ has features $e_{ij} \in \mathbb{R}^{D_e}$. Assume that we are also given a start node $v_s \in \mathcal{V}$ and a goal node $v_g \in \mathcal{V}$. At any stage of our search algorithm, we can partition the nodes of our graph into three sets as $\mathcal{V} = \text{CLOSE} \cup \text{OPEN} \cup \text{REST}$, where CLOSE are the nodes already explored, OPEN are candidate nodes for exploration (i.e., all nodes connected to any node in CLOSE, but not yet in CLOSE), and REST is the rest of the graph. Each *expansion* moves a node from OPEN to CLOSE, and adds the neighbors of the given node from REST to OPEN. We call the set of newly added fringe nodes $\mathcal{V}_{new}$ at each search step. At the start of the search procedure, CLOSE = $\{v_s\}$ and we expand the nodes until $v_g$ is encountered (i.e., until $v_g \in \text{CLOSE}$).

**Greedy best-first search**. We can perform *greedy best-first search* using a greedy fringe expansion policy, such that we always expand the node $v \in \text{OPEN}$ that minimizes $h(v, v_g)$. Here, $h : \mathcal{V} \times \mathcal{V} \longrightarrow \mathbb{R}$ is a tailored heuristic function for a given use case. In our work, we are interested in learning a function $h$ that predicts shortest path lengths, this way minimizing $|\text{CLOSE}|$ in a *greedy best-first search* regime.

**Imitation of perfect heuristics**. Partially observable Markov decision processes (POMDPs) are a suitable framework to describe the problem of learning search heuristics [1]. We can have $s = (\text{CLOSE}, \text{OPEN}, \text{REST})$ as our state, an action $a \in \mathcal{A}$ corresponds to moving a node from OPEN to CLOSE, and the observations $o \in \mathcal{O}$ are the features of newly included nodes in OPEN. Note that one could consider an MDP framework to learn heuristics, but the time complexity of operating on the whole state is in most cases prohibitive. We also define a history $\psi \in \Psi$ as a sequence of observations $\psi = o_1, o_2, o_3, \dots$. Our work leverages the observation that using a heuristic function during greedy best-first search that correctly determines the length of the shortest path between fringe nodes and the goal node will also yield minimal $|\text{CLOSE}|$. For training, we adopt a perfect heuristic $h^*$, similar to [1], which has full information about $s$ during search. Such oracle can provide ground-truth distances $h^*(s, v, v_g)$, where $v \in \text{OPEN}$. To conclude, we define a *greedy best-first search policy* $\pi_\theta$ that uses a parameterized heuristic $h_\theta$ to expand nodes from OPEN with minimal heuristic values. One could also directly use a POMDP solver for the above-described problem, but this approach is usually infeasible due to the dimensionality of the search state [19].

## 3 Related Work

**General purpose heuristic design**. There has been significant research in designing general-purpose heuristics for speeding up satisficing planning. The first set of approaches are based on simplifying the search problem for example using landmark heuristics [14, 16]. The next set of approaches aim to include novelty-based exploration in greedy best-first search [10–13]. The latter set of approaches

showed state-of-the-art performance (best-first width search [12, 13], BFWS) in numerous settings. We show that in domains where data is available, it can be more effective to incorporate a learned heuristic into a greedy best-first search procedure.

**Learning heuristic search**. There have been numerous previous works that attempt to learn search heuristics: Arfaee *et al.* [20] propose to improve heuristics iteratively, Virseda *et al.* [21] learn to combine heuristics to estimate graph node distances, Wilt *et al.* [22] and Garrett *et al.* [23] propose to learn node rankings, Thayer *et al.* [24] suggest to infer heuristics during a search, and Kim *et al.* [25] train a neural network to predict graph node distances. These methods generally do not consider the non-i.i.d. nature of heuristic search. Further, Bhardwaj *et al.* [1] propose SAIL, where heuristic learning is framed as an imitation learning problem with cost-to-go oracles. The SAIL heuristic uses hand-designed features tailored for obstacle avoidance, with a linear time-complexity in the number of explored grid nodes found to be colliding with an obstacle. Feature-engineering becomes more difficult as we attempt to learn heuristics on diverse graphs such as ones seen in Section 5.2, where we may need expert knowledge. Further, heuristics that do not have a constant time complexity in the size of the graph [1, 26–29] generally scale poorly with graph size and hence have constrained use cases. Recent approaches to learning heuristics include Retro* [2] by Chen *et al.*, where a heuristic is learned in the context of AND-OR search trees for chemical retrosynthetic planning. Our work focuses on a more general graph setting.

There has been significant progress on learning heuristics for NP-hard combinatorial optimization problems [30–32]. Focusing on solving NP-hard problems allows these approaches to design algorithms that are often non-exact and have a relatively large computational budget. This is not the case for methods that focus on polynomial time search, where learning-based methods are bounded by the determinism and time complexity of classical algorithms such as greedy best-first search.

**Learning general purpose search**. Learning general search policies is a very well-studied research area with a rich set of developments and applications. These include Monte Carlo Tree Search methods [33, 34], implicit planning methods [35–37], and imagination-based planning approaches [38, 39]. Learning search heuristics can be seen as a special case of *general purpose search*, where the search problem is treated as a partially observable Markov decision process with restricted action evaluation (see Section 4), and with models running in $\mathcal{O}(1)$ to remain competitive time-complexity-wise on problems where best-first search performs well. *General purpose search* methods do not take into account the above-mentioned constraints, which motivates the development of tailored approaches for learning heuristics [1, 2].

**Imitation learning**. Our approach builds on prior work in imitation learning (IL) with cost-to-go oracles. Cost-to-go oracles have been incorporated in the context of IL in methods such as SEARN [40], AggreVaTe [18], LOLS [41], AggrevaTeD [42], DART [43], and THOR [44]. SAIL [1] presents an AggreVaTe-based algorithm for learning heuristic search. We extend SAIL by incorporating a recurrent $Q$-like function, in which sense our algorithm more closely resembles AggreVaTeD by Sun *et al.* [42]. While a recurrent policy can be easily incorporated in AggreVaTeD, we cannot use a policy to evaluate actions. This is due to the fact that we would either have to evaluate all actions in a state, which is computationally infeasible, or we would have to give up on taking actions that are not in the most recent version of the search fringe, which would degrade the performance (see Section 4).

## 4 Path Heuristic with Imitation Learning

**Training objective**. With the aim of minimizing |CLOSE| after search, our goal is to train a parameterized heuristic function $h_\theta : \Psi \times \mathcal{V} \times \mathcal{V} \longrightarrow \mathbb{R}$ to predict ground-truth node distances $h^*$ and use $h_\theta$ within a greedy best-first policy $\pi_\theta$ at test time. More specifically, we assume access to a distribution over graphs $P_\mathcal{G}$, a start-goal node distribution $P_{v_{sg}}(\cdot \mid \mathcal{G})$, and a time horizon $T$. Moreover, we assume a joint state-history distribution $s, \psi \sim P_s(\cdot \mid \mathcal{G}, t, \pi_\theta, v_s, v_g)$, where $P_s$ represents the probability our search being in state $s$, at time $0 \leq t \leq T$ on graph $\mathcal{G}$ with pathfinding problem $(v_s, v_g)$, with a greedy best-first search policy $\pi_\theta$ using heuristic $h_\theta$. Hence, our goal can be summarized as minimizing the following objective:

---

**Algorithm 1:** PHIL— Sequential Heuristic Training

---

Obtain hyperparameters $T$, $\beta_0$, $N$, $m$, $t_\tau$;
Initialize $\mathcal{D} \leftarrow \emptyset$, $h_{\theta_1}$;
**for** $i = 1, \ldots, N$ **do**
    Sample $\mathcal{G} \sim P_{\mathcal{G}}$;
    Sample $v_s, v_g \sim P_{v_{sg}}$;
    Set $\beta \leftarrow \beta_0^i$;
    Set mixture policy $\pi_{mix} \leftarrow (1 - \beta) * \pi_{\theta_i} + \beta * \pi^*$;
    Collect $m$ trajectories $\tau_{ij}$ as follows;
    **for** $j = 1, \ldots, m$ **do**
        Sample $t \sim \mathcal{U}(0, ..., T - t_\tau)$;
        Roll-in $t$ time steps of $\pi_{\theta_i}$ to obtain $z_t$ and new state $s_t = (\text{CLOSE}^0, \text{OPEN}^0, \text{REST}^0)$;
        Roll-out trajectory $\tau_{ij}$ as follows;
        **for** $k = 1, \ldots, t_\tau$ **do**
            Update $s_{t+k-1}$ using $\pi_{mix}$ to get new state $s_{t+k}$ and new fringe state $\text{OPEN}^k$;
            Obtain new fringe nodes $\mathcal{V}_{new} = \text{OPEN}^k \setminus \text{OPEN}^{k-1}$;
            Update trajectory $\tau_{ij} \leftarrow \tau_{ij} \cup \{(\mathcal{V}_{new}, h^*(s_{t+k}, \mathcal{V}_{new}, v_g))\}$;
        Update dataset $\mathcal{D} \leftarrow \mathcal{D} \cup \{(\tau_{ij}, z_t)\}$ or $\mathcal{D} \cup \{(\tau_{ij}, 0)\}$;
    Train $h_{\theta_i}$ using TBTT on each $\tau \in \mathcal{D}$ to get $h_{\theta_{i+1}}$;
**return** *best performing $h_{\theta_i}$ on validation*;

---

$$\mathcal{L}(\theta) = \mathop{\mathbb{E}}_{\substack{\mathcal{G} \sim P_{\mathcal{G}}, \\ (v_s, v_g) \sim P_{v_{sg}} \\ t \sim \mathcal{U}(0, ..., T), \\ s, \psi \sim P_s}} \left[ \frac{1}{|\text{OPEN}|} \sum_{v \in \text{OPEN}} (h^*(s, v, v_g) - h_\theta(\psi, v, v_g))^2 \right] \tag{1}$$

Before we describe the algorithm that can be used to minimize $\mathcal{L}$, we rewrite $h_\theta$ to include a memory digest component ($z_t$), which represents an embedding of $\psi$ at time step $t$. Hence, $h_\theta$ becomes $h_\theta : \mathbb{R}^d \times \mathcal{O} \times \mathcal{V} \times \mathcal{V} \longrightarrow \mathbb{R}$, where $d$ is the dimensionality of our memory's embedding space. As opposed to previous methods [1], $z_t$ allows us to *automatically extract* relevant features for heuristic computations and concurrently *reduce the computational complexity* of the heuristic function. Further, as shown in [1], if we would use $h_\theta$ to evaluate all actions in a state (i.e., recalculate the heuristic values of all nodes in OPEN), we would need a squared reduction in the number of expanded nodes compared with BFS for PHIL to bring performance benefits over BFS, which however may not be possible on all datasets. Hence, we constrain the heuristic only to evaluate new OPEN nodes which we obtain after moving a node to CLOSE, calling the set of new fringe nodes $\mathcal{V}_{new}$ after each expansion. In practice, the policy $\pi_\theta$ yields an algorithm equivalent to greedy best-first search, with the heuristic function replaced by $h_\theta$.

## 4.1 Learning algorithm & architecture

**Imitation learning algorithm**. In Algorithm 1, we present the pseudo-code of the IL algorithm used to train our heuristic models (Figure 3). The high-level idea of our algorithm is that we aggregate trajectories of search traces (i.e., sequences of new fringe nodes) and use truncated backpropagation through time to optimize $h_\theta$ after each data-collection step. In particular, after sampling a graph $\mathcal{G}$ and a search problem $v_s, v_g$, we use our greedy learned policy $\pi_\theta$ induced by $h_\theta$ to roll-in for $t \sim \mathcal{U}(0, \ldots, T - t_\tau)$ expansions, where $T$ is the episode time horizon, and $t_\tau$ is the roll-out length. From our roll-in, we obtain a new state $s = (\text{CLOSE}^0, \text{OPEN}^0, \text{REST}^0)$, and an initial memory state $z_t$. After our roll-in, we roll-out for $t_\tau$ steps using our mixture policy $\pi_{mix}$, which is obtained by probabilistically blending $\pi_\theta$ and the greedy best-first policy induced by the oracle heuristic $\pi^*$. In a roll-out, we collect sequences of new fringe nodes, together with their ground-truth distances to the goal $v_g$, given by $h^*$. Once the roll-out is complete, we append the obtained trajectory and the initial state for the following optimization using backpropagation through time. Further analysis on the trade-offs between using rolled-in states $z_t$ or zeroed-out states for training can be found in the supplementary material.

Note that we could also use supervised learning-based approaches to sample a fixed dataset of $(v_s, v_g, h^*(s, v_s, v_g))$ 3-tuples and train a model to predict node distances conditioned on their features. However, our experiments in Section 5 demonstrate that ignoring the non-i.i.d. nature of heuristic search negatively impacts model performance, with supervised learning-based methods performing up to $40\times$ worse.

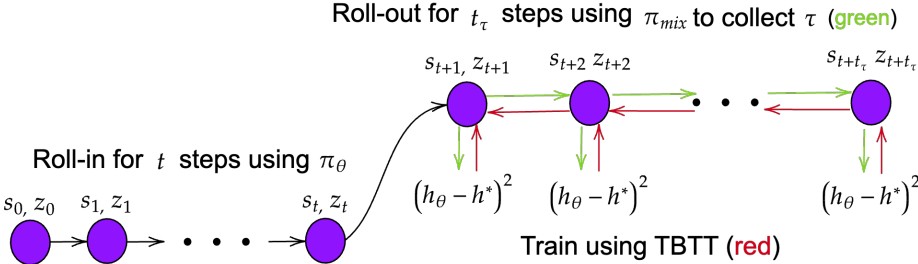

**Figure 3:** This figure demonstrates the core idea behind our IL algorithm. We present the roll-in phase on the left-hand side, where our policy is rolled in for $t$ steps to obtain state $s_t$ and embedding $z_t$. On the right-hand side, we show the trajectory collection and training steps, where we aggregate the trajectory for downstream training (green) and use truncated backpropagation through time on the collected dataset (red).

**Recurrent GNN architecture.** In each forward pass, $h_\theta$ obtains a set of new fringe nodes $\mathcal{V}_{new}$, the goal node $v_g$, and the memory $z_t$ at time step $t$. We represent each node in $\mathcal{V}_{new}$ using its features $x_i \in \mathbb{R}^{D_v}$, and likewise the goal node $v_g$ using its features $x_g \in \mathbb{R}^{D_v}$. Further, for each $i \in \mathcal{V}_{new}$, we uniformly sample an $n \in \mathbb{N}_{\geq 0}$ bounded set of nodes present in the 1-hop neighborhood of $i$, calling this set $\mathcal{N}_i$, with $|\mathcal{N}_i| \leq n$. This sampling step produces a set of neighboring node features, where each $j \in \mathcal{N}_i$ has features $x_j \in \mathbb{R}^{D_v}$, and corresponding edge features $e_{ij} \in \mathbb{R}^{D_e}$.

---

**Algorithm 2:** Heuristic func. ($h_\theta$) forward pass

Obtain $x_i, x_j (j \in \mathcal{N}_i), e_{ij}, x_g z_t$;
$x_i \leftarrow f(x_i, x_g, D_{EUC}(x_i, x_g), D_{COS}(x_i, x_g))$;
$x_j \leftarrow f(x_j, x_g, D_{EUC}(x_j, x_g), D_{COS}(x_j, x_g))$;
$g_i \leftarrow \phi(x_i, \oplus_{j \in \mathcal{N}_i} \gamma(x_i, x_j, e_{ij}))$;
$g_i', z_{i,t+1} \leftarrow \text{GRU}(g_i, z_t)$;
$z_{t+1} \leftarrow \overline{z_{i,t+1}}$;
$\hat{h}_i \leftarrow \text{MLP}(g_i', x_g)$;
**return** $\hat{h}_i, z_{t+1}$;

---

$h_\theta$ **forward pass.** Algorithm 2 presents a single forward pass of $h_\theta$. The forward pass outputs predicted distances of the new fringe nodes to the goal $\hat{h}_i$, together with an updated memory digest $z_{t+1}$. In Algorithm 2, $f, \phi, \gamma$, GRU[45], MLP are each parameterised differentiable functions, with $\phi, \gamma$ representing the *update* and *message* functions [46] of a graph neural network, respectively.

In our forward pass, using the function $f$, we first project $x_i, x_j$ into a node embedding space, together with the goal features $x_g$, and their Euclidean ($D_{EUC}$) and cosine distances ($D_{COS}$). After that, using a 1-layer GNN, we perform a single convolution over each $x_i$ and the corresponding neighborhood $\mathcal{N}_i$, to obtain $g_i$. The specific GNN choice is a design decision left to the practitioner, and further analysis of GNN choices can be found in Appendix D. Our graph convolution processing step allows us to easily incorporate edge features and work with variable sizes of $\mathcal{N}_i$. After the graph convolution, we apply the GRU module over each embedding $g_i$ to obtain hidden states $z_{i,t+1}$, and new embeddings $g_i'$. We compute the sample mean of $z_{i,t+1}$ for each node $i \in \mathcal{V}_{new}$ to obtain a new hidden state $z_{t+1}$, and process $g_i'$ with $x_g$ using an MLP to compute the distances between the graph nodes.

The intuitive explanation for using history embeddings is to address the partial observability of the problem. PHIL does not have access to the full graph because of time complexity concerns. The history embeddings provide a mechanism for keeping track of the belief over the full state of the graph during search. Further, the GNN allows for easily incorporating local neighborhoods and edge features. Please refer to Appendix D where we discuss the effects of our design choices.

**Permutation invariant $\mathcal{V}_{new}$ embedding.** There is a trade-off between processing new fringe nodes in batch, as in Algorithm 2, and processing them sequentially. Namely, when we process the nodes in batch, we do not use the in-batch observations to predict batch node values, which means that $z_t$ is slightly outdated. On the other hand, in PHIL, batch processing allows us to compute the heuristic values of all $v \in \mathcal{V}_{new}$ in parallel on a GPU and preserves the memory's permutation invariance with respect to nodes in $\mathcal{V}_{new}$. That is, because our observations are nodes & edges of a graph, the respective observation ordering usually does not contain inductive biases useful for predictions, which means that we can apply a permutation invariant operator such as the mean of all new states $z_{i,t+1}$ to obtain an aggregated updated state. This approach provides additional scalability as we can process values in parallel and PHIL does not have to infer permutation invariance in $\mathcal{V}_{new}$ from data.

**Runtime complexity**. Since $\forall i \in \mathcal{V}_{new} : |\mathcal{N}_i| \leq n$, Algorithm 2 together with neighborhood sampling runs in up to $nc_1 + (n + 1)c_2$ operations per each node $i \in \mathcal{V}_{new}$, which is $\mathcal{O}(1)$ with respect to the size of the graph. Here, $c_1$ is the maximal number of operations associated with evaluating a node, such as performing robot collision checks in dynamically constructed graphs, and $c_2$ is the maximal count of total model operations (e.g., $f$ & $\gamma$ operations) on the node set $\{i\} \cup \mathcal{N}_i$. Note that for this analysis, we assume that $c_1$ is bounded. In general, we expect to learn a better search heuristic with increasing $n$, but in some use cases, $c_1$ may dominate overall complexity, which means the hyperparameter $n$ is helpful for practitioners to tune trade-offs between constant factors and search effort minimization.

## 5   Experiments

In our experiments, we evaluate PHIL both on benchmark heuristic learning datasets [1] (Section 5.1) as well on a diverse set of graph datasets (Section 5.2). Finally, we show that PHIL can be applied to efficient planning in the context of drone flight (Section 5.3). Our main goal is to assess how PHIL compares to baseline methods in terms of necessary expansions before the goal node is reached. Please refer to the supplementary material for information about baselines, an ablation study, and additional experiment details.

### 5.1   Heuristic search in grids

In Section 5.1, we evaluate PHIL on 8, $200 \times 200$ 8-connected grid graph-based datasets by Bhardwaj *et al.* [1]. These datasets present challenging obstacle configurations for naive greedy planning heuristics, especially when $v_s$ is in the bottom-left of the grid, and $v_g$ in the top-right. Each dataset contains 200 training graphs, 70 validation graphs, and 100 test graphs. Example graphs from each dataset can be found in Table 1.

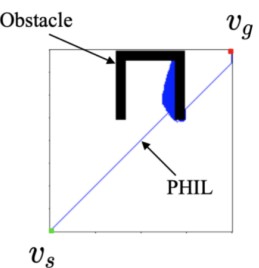

We train PHIL with a hyperparameter configuration of $T = 128$, $t_\tau = 32$, $\beta_0 = 0.7$, $n = 8$, and using rolled-in $z_t$ states as initial states for training. We use a 3-layer MLP of width 128 with *LeakyReLU* activations, followed by a DeeperGCN [47] graph convolution with *softmax* aggregation. Our memory's embedding dimensionality is 64. The node features are 2D grid coordinates. See

**Figure 4:** Example of PHIL escaping local search minima.

Appendix C for an overview of our baselines and datasets. Note that although using positional representations [48] might be useful for this search problem, it is not scalable as the graph size increases (we refer the reader to Appendix A for more details).

**Discussion.** As we can see in Table 1, PHIL outperforms the best baseline (SAIL) on all datasets, with an average reduction of explored nodes before $v_g$ is found of $58.5\%$. Qualitatively, observing Figure 5, we can attribute these results to PHIL's ability to **reduce the redundancy in explored nodes** during a search. Further, PHIL is also **capable of escaping local minima**, which is illustrated in Figure 4. However, note that we occasionally observe failure cases in practice, where PHIL gets stuck in a bug trap-like structure. We discuss possible remedies and opportunities for future work in the supplementary material.

**Runtime & convergence speed**. PHIL converges in up to $N = 36$ iterations, with $m = 1, t_\tau = 32$ (i.e., after observing less than $N * t_\tau * max(|\mathcal{V}_{new}|) \approx 9,216$ shortest path distances, where we

| Dataset | Graph Examples | SAIL | SL | CEM | QL | $h_{euc}$ | $h_{man}$ | A* | MHA* | BFWS | Neural A* | PHIL |
|---|---|---|---|---|---|---|---|---|---|---|---|---|
| Alternating gaps | | 0.039 | 0.432 | 0.042 | 1.000 | 1.000 | 1.000 | 1.000 | 1.000 | 0.34 | 0.546 | **0.024** |
| Single Bugtrap | | 0.158 | 0.214 | 0.057 | 1.000 | 0.184 | 0.192 | 1.000 | 0.286 | 0.099 | 0.394 | **0.077** |
| Shifting gaps | | 0.104 | 0.464 | 1.000 | 1.000 | 0.506 | 0.589 | 1.000 | 0.804 | 0.206 | 0.563 | **0.027** |
| Forest | | 0.036 | 0.043 | 0.048 | 0.121 | 0.041 | 0.043 | 1.000 | 0.075 | 0.039 | 0.399 | **0.027** |
| Bugtrap+Forest | | 0.147 | 0.384 | 0.182 | 1.000 | 0.410 | 0.337 | 1.000 | 3.177 | 0.149 | 0.651 | **0.135** |
| Gaps+Forest | | 0.221 | 1.000 | 1.000 | 1.000 | 1.000 | 1.000 | 1.000 | 1.000 | 0.401 | 0.580 | **0.039** |
| Mazes | | 0.103 | 0.238 | 0.479 | 0.399 | 0.185 | 0.171 | 1.000 | 0.279 | 0.095 | 1.000 | **0.069** |
| Multiple Bugtraps | | 0.479 | 0.480 | 1.000 | 0.835 | 0.648 | 0.617 | 1.000 | 0.876 | 0.169 | 0.331 | **0.136** |

**Table 1:** The number of expanded graph nodes of PHIL with respect to SAIL. We can observe that out of all baselines, SAIL performs best. PHIL outperforms SAIL by $58.5\%$ on average over all datasets, with a maximal search effort reduction of $82.3\%$ in the *Gaps+Forest* dataset.

take $max(|\mathcal{V}_{new}|) = 8$ as the maximal size of $\mathcal{V}_{new}$). According to figures reported in [1], this is approximately $5\times$ less data than it takes for SAIL to converge.

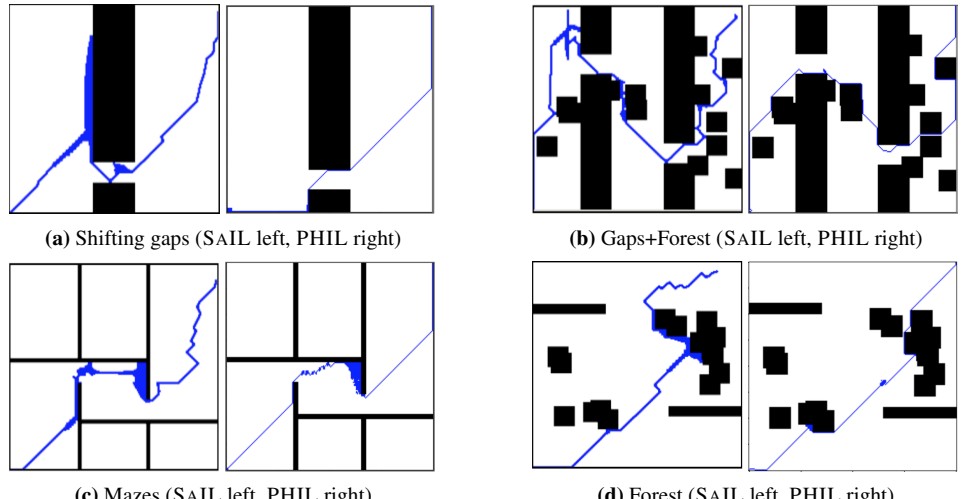

(a) Shifting gaps (SAIL left, PHIL right)          (b) Gaps+Forest (SAIL left, PHIL right)

(c) Mazes (SAIL left, PHIL right)          (d) Forest (SAIL left, PHIL right)

**Figure 5:** In each image pair of this figure, we provide a qualitative comparison with the SAIL method. In particular, we show comparisons on the *Shifting gaps*, *Gaps+Forest*, *Mazes*, and *Forest* datasets. We can observe that PHIL (right) learns the appropriate heuristics for the given dataset and makes fewer redundant expansions than SAIL (left).

## 5.2 Search in real-life graphs of different structures

In this experiment, our goal is to demonstrate the general applicability of PHIL to various graphs. We train PHIL on 4 different groups of graph datasets: citation networks, biological networks, abstract syntax trees (ASTs), and road networks. We use the same graph for citation networks and road networks for training and evaluation, and we use 100 random $v_s, v_g$ pairs for testing. In the case of biological networks and ASTs, we usually have train/validation/test splits of $80/10/10$, and in the case of the OGB [49] datasets, we use the provided splits.

Similarly as in Section 5.1, our MLP has four layers of width 128 with *LeakyReLU* activations and we use a DeeperGCN [47] graph convolution with *softmax* aggregation. The utilized node and edge features are the provided features in each dataset, except for a few minor modifications which are discussed in Appendix A & Appendix C. We train an MLP of depth 5 and width 256 using supervised learning (SL) for our learning-based baseline method.

**Discussion**. The results presented in Table 2 suggest that PHIL can learn superior search heuristics compared with baseline methods, outperforming top baselines per dataset in terms of visited nodes

| | Dataset | $|\mathcal{D}|$ | $|\bar{\mathcal{V}}|$ | $|\bar{\mathcal{E}}|$ | SL | A* | $h_{euc}$ | BFS | SAIL | BFWS | PHIL |
|---|---|---|---|---|---|---|---|---|---|---|---|
| Citation Networks | Cora (Sen *et al.* [50]) | 1 | 2,708 | 5,429 | 2.201 | 2.067 | 1.000 | 4.001 | 0.669 | 1.378 | **0.475** |
| | PubMed (Sen *et al.* [50])) | 1 | 19,717 | 44,338 | 2.157 | 2.983 | 1.000 | 3.853 | 1.196 | 1.000 | **0.745** |
| | CiteSeer (Sen *et al.* [50])) | 1 | 3,327 | 4,732 | 1.636 | 1.487 | 1.000 | 2.190 | 1.062 | 0.951 | **0.599** |
| | Coauthor (cs) (Schur *et al.* [51]) | 1 | 18,333 | 81,894 | 1.571 | 1.069 | 1.000 | 2.820 | 1.941 | 1.026 | **0.835** |
| | Coauthor (physics) (Schur *et al.* [51]) | 1 | 34,493 | 247,962 | 4.076 | 1.081 | 1.000 | 4.523 | – | 1.012 | **0.964** |
| Biological Networks | OGBG-Molhiv (Hu *et al.* [49]) | 41,127 | 25.5 | 27.5 | 1.086 | 1.065 | **1.000** | 1.267 | 1.104 | 1.146 | 1.016 |
| | PPI (Zitnik *et al.* [52]) | 24 | 2,372.67 | 34,113.16 | 0.772 | 0.831 | 1.000 | 5.618 | 1.746 | 3.941 | **0.658** |
| | Proteins (Full) (Morris *et al.* [53]) | 1,113 | 39.06 | 72.82 | 0.995 | 0.997 | 1.000 | 2.645 | 0.891 | 0.966 | **0.831** |
| | Enzymes (Morris *et al.* [53]) | 600 | 32.63 | 62.14 | 1.073 | 1.007 | 1.000 | 1.358 | 1.036 | 0.992 | **0.757** |
| ASTs | OGBG-Code2 (Hu *et al.* [49]) | 452,741 | 125.2 | 124.2 | 1.196 | 1.013 | 1.000 | 1.267 | 1.029 | **0.817** | 1.219 |
| Road Networks | OSMnx - Modena (Boeing [54]) | 1 | 29,324 | 38,309 | 2.904 | 3.085 | 1.000 | 3.493 | 1.182 | 0.997 | **0.489** |
| | OSMnx - New York (Boeing [54]) | 1 | 54,128 | 89,618 | 39.424 | 36.529 | 1.000 | 63.352 | 1.583 | 1.013 | **0.962** |

**Table 2:** Comparison of PHIL with baseline approaches on $4$ groups of datasets: citation networks, biological networks, abstract syntax trees, and road networks. "$-$" denotes being out of a 4-day's training time limit. We can observe that, on average across all datasets, PHIL outperforms the best baseline per dataset by $13.4\%$. Discounting the OGBG datasets, this number becomes $19.5\%$.

during a search by $13.4\%$ on average. Two datasets where PHIL fell short compared to other baselines are the *OGBG-Molhiv* and *OGBG-Code2* datasets. The *OGBG-Code2* dataset adopts a *project split* [55] and OGBG-Mohliv adopts a *scaffold split* [56], both of which ensure that graphs of different structure are present in the training & test sets. Although PHIL improved upon uninformed search (BFS) in the OGB datasets, structural graph consistency is explicitly discouraged in the above-mentioned OGBG splits. Without the OGBG datasets, PHIL improves on the top baselines per dataset by $19.5\%$ on average, and upon the Euclidean node feature heuristic ($h_{euc}$) by $20.4\%$. Note that we trained PHIL up to $N = 60$ iterations, which means that it only encountered a small subset of the pathfinding problems in the single graph setting, which means that PHIL had to generalize to learn useful heuristics. Even in Cora, the $|\mathcal{D}| = 1$ dataset with least number of nodes, PHIL observed roughly $6,000$ node distances during training, which is less than $0.2\%$ of total distances in the Cora graph.

## 5.3 Planning for drone flight

In our final experiment, we use PHIL to plan collision-free paths in a practical drone flight use case within an indoor environment. We built our environment using the CoppeliaSim simulator [57], and the Ivy framework [58]. Figure 6 presents the environment which we refer to as *room adversarial* in Table 3. For more detail about each environment, please refer to the supplementary material. We discretize the environments into 3D grid graphs of size $50 \times 50 \times 25$, and randomly remove 5 sub-graphs of size $5 \times 5 \times 5$ both during training and testing, this way simulating real-life planning scenarios with random obstacles. The hyperparameter configuration and the specific architecture we utilize are equivalent to Section 5.1, but with $n = 4$. Likewise, the node features are 3D grid coordinates, and the baselines include supervised learning (SL), $h_{euc}$, A*, and BFS, similarly as in Sections 5.1, 5.2. In Table 3 we report the ratio of expanded nodes with respect to $h_{euc}$.

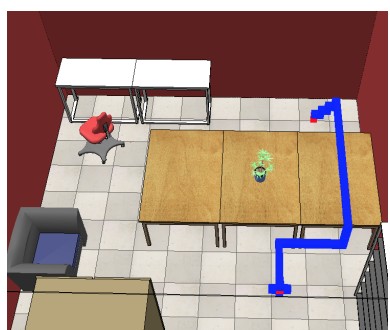

**Figure 6:** This figure illustrates the *room adversarial* environment with an example planning problem (red) and the expanded graph by PHIL (blue).

**Video demo**. We provide a video demonstration of PHIL running in *room adversarial*: https://cutt.ly/eniu5ax.

| Dataset | SL | A* | $h_{euc}$ | BFS | SAIL | BFWS | PHIL | Shortest path |
|---|---|---|---|---|---|---|---|---|
| Room simple | 1.124 | 76.052 | 1.000 | 291.888 | 0.973 | 1.286 | **0.785** | 0.782 |
| Room adversarial | 2.022 | 67.215 | 1.000 | 238.768 | 0.944 | 1.583 | **0.895** | 0.853 |

**Table 3:** Results of PHIL in the context of planning for indoor UAV flight. PHIL outperforms all baselines in both the *room simple* and *room adversarial* environments while remaining close performance-wise to the optimal number of expansions.

**Discussion**. As we can observe in Table 3, PHIL outperforms all baselines in both environments. Interestingly, PHIL expands only approximately $0.3\%$ more nodes in the simple room than least possible and $4.9\%$ more in the adversarial room case. The same figures for the greedy method ($h_{euc}$) are $27.8\%$ and $17.2\%$, respectively. These results indicate that PHIL is capable of learning planning strategies that are close to optimal in both *simple* and *adversarial* graphs, while the performance of naive heuristics degrades.

### 5.4 Runtime Analysis

We summarize test run-times of different approaches in Appendix G. PHIL runs $57.9\%$ faster than BFWS and $32.2\%$ faster than SAIL, and not much slower than traditional A* ($34.7\%$) and $h_{man}$ ($18.3\%$). Although Neural A* is $71.0\%$ faster than PHIL due to the fact that it casts the whole search process into matrix operations on images, it cannot be employed in a generic search setting.

## 6 Conclusion

We consider the problem of learning to search for feasible paths in graphs efficiently. We propose a model and a training procedure to learn search heuristics that can be easily deployed across diverse graphs, with tuneable trade-off parameters between constant factors and performance. Our results demonstrate that PHIL outperforms current state-of-the-art approaches and can be applied to various graphs with practical use cases.

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
