# OpenReview forum: "Learning Graph Search Heuristics"
_logconference.io/LOG/2022/Conference — LoG 2022 Poster_

### Official Review · Reviewer_BUfS · 2022-10-13

**Overall Score:** 8
**Confidence:** 4

**Review:**

The authors study the fundamental problems of searching trajectories between two nodes. They propose a novel pathfinding method by exploiting inductive biases of the graph representations. Specifically, they adopt imitation learning to find path heuristics with a learnable explored graph memory. Finally, they conduct extensive experiments to show the effectiveness and efficiency of the proposed method.

Overall speaking, I found that the idea of this paper is interesting and the method itself is easy to follow. The paper is well-written and seems to be of significant contribution to this research area. The proposed method runs in constant time independent of graph sizes and is much more efficient compared with conventional search algorithms and hand-craft heuristics.

Through your experiments, we can find that PHIL incorporates graph representations to reduce the redundancy in explored nodes during the graph search. It means that PHIL exploits some important local node structures, and these critical nodes are targeted to the goal node with the shortest path distances. Could you provide more visualizations on these critical nodes in the search path? It would be interesting to how PHIL works when incorporating graph structures.

---

### Official Review · Reviewer_yPEZ · 2022-10-22

**Overall Score:** 6
**Confidence:** 4

**Review:**

Summary:

The paper studies the well-studied problem of finding a path between two nodes. While being a fundamental problem, this has a range of applications in various fields such as robotics and graphs. The paper proposes a training algorithm for discovering paths between two nodes based on graph representation learning and imitation learning. The results show that the learned representations are useful and can be efficient in practice even with large graphs as the running time does not depend on the graph size.

Reasons for score:

I like the overall idea of the paper. The use of imitation learning is sort of novel in this context. The experimental results are also strong. The proposed use case is also interesting. My major concern is regarding the clarity on the differences between the existing techniques and the proposed one along with some other ones listed below. Hope the comments will help in improving the paper. I am willing to improve my score after the concerns are addressed in the rebuttal.

Positives:

The paper studies a fundamental problem and proposes an unusual approach to solving it. The proposed idea of using imitation learning to solve the underlying combinatorial problem is interesting. The experimental results are strong. More specifically, Table 1 shows that the proposed method works outperform the baselines. Section 5.3 is really nice. This kind of experiment is uncommon, especially for graph combinatorial problems and I really appreciate it.

Comments/Concerns:

C1. The paper does not describe why the existing techniques on combinatorial problems won't be useful. There are sophisticated approaches [2] to solving NP-hard problems (even complex problems such as influence maximization [3]). Here the studied problem is not even NP-hard. This part is a little confusing. The paper mentions briefly (refer to Section 3, third paragraph) that RL approaches won't work but the discussion needs to be substantial. It would help the community, especially the one that looks at solving the graph combinatorial problems via a learning framework.

C2. The BFS problem has been well tackled by the Graph500 benchmark [4]. It would be good to show some results against the top algorithms in the leaderboard of the Graph500 benchmark.

C3. There is work on positional representation (embeddings) of nodes [1]. It seems for this particular search problem, they might be useful. A discussion would be helpful. As a follow-up, what kind of initial features are being used in PHIL? How are they helping (intuitively)?

C4. The difference between the proposed method, PHIL, and the existing ones (SAIL and AggreVate) needs some clarity. Why the trivial extension (modification) of these two can't perform well? What is PHIL's advantage over AggreVate in this particular context?

C5. The complexity analysis is not rigorous. In particular, can c_1 and c_2 be bounded even for special cases?

C6. It is not clear if the proposed method is inductive. In the experiments with the real graphs, the method has been trained and evaluated on the same graph (Sec 5.2).

[1] You, Jiaxuan, Rex Ying, and Jure Leskovec. "Position-aware graph neural networks." In International conference on machine learning, pp. 7134-7143. PMLR, 2019.

  [2]  Khalil, Elias, Hanjun Dai, Yuyu Zhang, Bistra Dilkina, and Le Song. "Learning combinatorial optimization algorithms over graphs." Advances in neural information processing systems 30 (2017).

 [3] Manchanda, Sahil, Akash Mittal, Anuj Dhawan, Sourav Medya, Sayan Ranu, and Ambuj Singh. "Gcomb: Learning budget-constrained combinatorial algorithms over billion-sized graphs." Advances in Neural Information Processing Systems 33 (2020): 20000-20011.

[4] https://graph500.org/

---

### Official Review · Reviewer_Wez5 · 2022-10-22

**Overall Score:** 6
**Confidence:** 3

**Review:**

This paper studies the problem of searching for a path between two nodes in a graph, and the goal is to minimize the number of searched nodes before reaching the target. They propose an imitation learning
framework with a heuristic function to predict the distance between the fringe and target nodes. They incorporate the graph neural network with the one-hop neighbour to encode the fringe nodes and apply GRU to represent the searched history as a state. They conduct extensive experiments on three datasets and demonstrate the superior performance of their framework.

## Rating
I recommend an acceptance for this paper. Despite some concerns (c.f. below), the idea is novel and the results are strong.

## Pros
* The overall design of the imitation learning framework and the heuristic function is novel. The graph neural network effectively enriches the representation of the fringe nodes.
* The experiment demonstrates significant improvement over other baselines. Besides, they provided open-sourced code from the supplementary material.

## Concerns
* Although all the necessary details are provided, it might be better to intuitively explain some critical design choices, e.g., why GNN can improve the performance in Algorithm 2, why the learned history embedding can enhance the results, etc. Some ablation studies on these critical designs might be interesting.
* Prediction is based on the node features. What if node features do not have a strong correlation w.r.t. the distance between fringe node and the target node?
* In Algorithm 2, why not consider taking the z_{i, t+1} with i  decided by the trajectory as z_{t+1} rather than taking the mean of all the fringe nodes? If we take z_{t+1} with the actuarial chosen node, we can add z_i in the final MLP to include the history information in the final prediction. This design seems more natural than integrating all the embeddings of the fringes nodes into the history.

## Minor issues
* The paper claims a constant time independent of graph sizes, which might be misleading since it seems that the run time for each step is constant, but the overall run time depends on the length of the shortest path.
* Typo in Line 65: Fdomains -> domains.
* Notation issue in (1):  OPEN should be with some subscript to indicate the dependence w.r.t. G, vs, vg and s.
* In Algorithm 2, Line 3, j should be anyone from the sampled set of 1 hop neighbour rather than a single j.

---

### Meta-Review · Area_Chair_twQj · 2022-11-14

**Confidence:** 4
**Recommendation:** Accept

**Meta Review:**

This work proposes Path Heuristic with Imitation Learning (PHIL), a neural method for pathfinding on graphs.  PHIL uses graph representation learning and imitation learning, and experiments show promising reductions in terms of explored nodes compared to baseline methods on average.

Reviewers favorably assessed this work, with a few caveats and limitations appearing during the discussion period:

- Reviewer Wez5 mentions a few design choices are confusing and not well justified; authors clarify some of these experiments are evident in the appendix already.

- Reviewer yPEZ appreciates the novel ideas and strong results, but takes issue with a benchmarks that may not sufficiently differentiated against, and whose shortcomings should be made clear.  The authors' response covers some of these points, but it is encouraged that this discussion makes it into the final version of the paper.

- Reviewer BUfs assesses the paper clearly positively, but wishes for clarification about the role of critical nodes and their visualization; authors respond with some visualizations in the Appendix.

*Strengths*

- S1. The proposed idea is novel.
- S2. The idea shows strong empirical performance.

*Weaknesses*

- W1. Some design choices are not clearly rationalized.
- W2. The shortcomings of related work and differentiation could be improved.

S2 > S1 > W2 > W1 is my train of thought for the final assessment.

I encourage the authors to incorporate some of the back and forth clarifications, especially with Reviewers yPEZ and Wez5, into the final version of the paper to avoid these same confusions for future readers.

---

### Decision · Program_Chairs · 2022-11-22

Accept (Poster)